# Design of and Experiment with Secondary Cutting Equipment for Broccoli

**Jiangming Jia** [1,2], **Runze Hu** [1], **Liqun Chen** [3], **Tianlong Chen** [1] **and Jianneng Chen** [1,2,*]

1 Faculty of Mechanical Engineering & Automation, Zhejiang Sci-Tech University, Hangzhou 310018, China; jarky@zstu.edu.cn (J.J.); 202030605188@mails.zstu.edu.cn (R.H.); 201820501006@mails.zstu.edu.cn (T.C.)
2 Key Laboratory of Transplanting Equipment and Technology of Zhejiang Province, Hangzhou 310018, China
3 Mechatronics and Automotive Engineering College, Huzhou Vocational &Technical College, Huzhou 313099, China; 2019035@huvtc.edu.cn
* Correspondence: jiannengchen@zstu.edu.cn; Tel.: +86-130-6570-1536

**Abstract:** To solve the problem of large-sized blocks in single-process broccoli cutting, this paper proposes the design of broccoli secondary cutting equipment, in which the screening device with differential round belts, spiral attitude-adjusting mechanism, double-baffle conveyor belt, block-centering chute and disc-type cutter are analyzed and designed. According to the simulation of the motion of the blocks on the differential belts, the speeds of the two belts were 300 mm/s and 600 mm/s, respectively. The kinematic analysis of the spiral attitude-adjusting mechanism was performed, and the speed of the spiral rod was calculated to be greater than 64.5 rpm. The speed of the double-baffle conveyor belt was greater than 10.61 rpm to not obstruct the blocks and achieve diversion. A force analysis of the inclined centering chute was performed to reduce the damage during block conveying, and the inclination angle of the inclined centering chute was calculated to be greater than 27.02°. The parameters of the blade and its driving motor were calculated. Effective secondary cutting equipment for broccoli was thus developed. After secondary cutting, the size difference of two small florets obtained was between 0–8 mm, the success rate of cutting was 94.8%, and the efficiency was 47 pieces/min, which verified the reasonableness and feasibility of the second cutting equipment scheme.

**Keywords:** broccoli florets; second cutting; screening device; spiral adjustment; double-baffle conveyor belt; centering chute

## 1. Introduction

Broccoli is a highly sought-after vegetable due to its high nutritional values, including fiber, protein, lipids, vitamins, and minerals. In addition, its positive impact on human health can be attributed to its high content of phytochemicals, such as flavonoids and thioglucosides [1–4]. Therefore, global broccoli production continues to grow, and in 2017, global production exceeded 26 million tons, of which China and India were the world's major broccoli producers [5,6]. Consumers must cut and clean broccoli that is primarily sold as whole plants. In addition, broccoli, which has a large rhizome that is not typically eaten, causes a large amount of household waste. Therefore, selling chopped broccoli directly will save consumers' time and retain the rhizome of broccoli to reduce pollution. Particularly for broccoli exports, the size of florets must be less than 70 mm in China [7,8]. Crop products are different from industrial products in that each object is complex and variable. Although having specific properties, each individual has its special characteristics. In addition, compared to tomatoes, bell peppers, potatoes, and other regular shaped vegetables, broccoli has a special umbrella-like shape, disorganized internal forks, thick main stems, and fragile surface pistils, all of which make broccoli processing and storage more difficult. There is relatively little academic research in this direction [9–16]. They can be roughly divided into two categories by tool type: equipment with spherical tools and

equipment with cylindrical and cross tools. They have their advantages and disadvantages, respectively: On one hand, spherical cutter equipment takes Liqun Chen's Cutting and Throwing Head Mechanism of Broccoli as an example[13], the broccoli is manually placed on a tray and after a series of drive chain drives including incomplete gears, the chainplate moves to the intermittent stage when the cylinder pushes the connecting rod to drive the spherical cutter to cut and the rhizome falls inside the cutter. After the return stroke of the cylinder, the rootstock is thrown out. Therefore, the spherical cutter equipment has a similar shape to broccoli, with a more regular circular cross-section, but the cutter must have an intermittent opening and closing action, while the individual branches of the broccoli rhizome have different diameters, and the size of the blocks is not uniform after one cut, and there are many larger size blocks. On the other hand, cylindrical and cross cutter equipment takes D-CORE 50i/30i Auto Corer as an example[18]. The worker places the broccoli on the pallet of the conveyor belt, and the cylindrical cutter is used to core the broccoli from the top down, and then the cross-shaped cutter is used to cut the broccoli into equal pieces. Obviously, this method can be divided into four parts of the same size, but the cross-sections of the final blocks are worse than those of the spherical cutter and inevitably there are blocks cut in half. Rhizome removed by the first cylindrical cutter block contains usable parts that are bound to be wasted. Thus it does not match the biological characteristics of broccoli-like spheres, moreover, the two cutting actions significantly reduce efficiency.

From the above, it is clear that among the two types of equipment, the spherical tool equipment is more suitable for the biological characteristics of broccoli to obtain better blocks, while all the equipment has a common disadvantage that the size of the blocks obtained from a single cut varies greatly and many do not meet the processing requirements. Therefore, there is an urgent need for a method to screen the florets obtained from one cutting and to cut them twice to obtain the required florets.

In this paper, we propose, develop and test screening and secondary cutting equipment for blocks larger than 70 mm after primary cutting. We have simulated the motion of crossed blocks on the differential round belt and theoretically analyzed the attitude adjustment process of the blocks in the double-spiral adjusting mechanism, and calculated and designed the double-baffle conveyor belt, centering chute, and disc-type cutter. With this equipment for block-cutting test, the blocks which are obtained from primary cutting are screened, adjusted attitude, conveyed, and secondary cutting. The developed equipment achieves a high success rate to obtain small florets with better uniformity that meets market requirements. The broccoli secondary cutting equipment proposed in this paper not only solves practical engineering problems but also has important significance in academic aspects. The simulation calculation ideas mentioned in this paper can also be applied to other crops, such as solving processing problems such as umbrella-shaped mushrooms and cabbage with thick main stems.

## 2. Materials and Method

### 2.1. Materials

The broccoli variety considered in this study was "Zheqing 95", which was selected by the Zhejiang Academy of Agricultural Sciences and planted and grown in Wenling, Taizhou, and Ningbo, Zhejiang Province. As shown in Figure 1, the components of broccoli include large stems, medium stems, small stems, and buds. The buds of "Zheqing 95" were measured to be between 140 and 160 mm in diameter (D) and between 70 and 90 mm in height (H) [9,11].

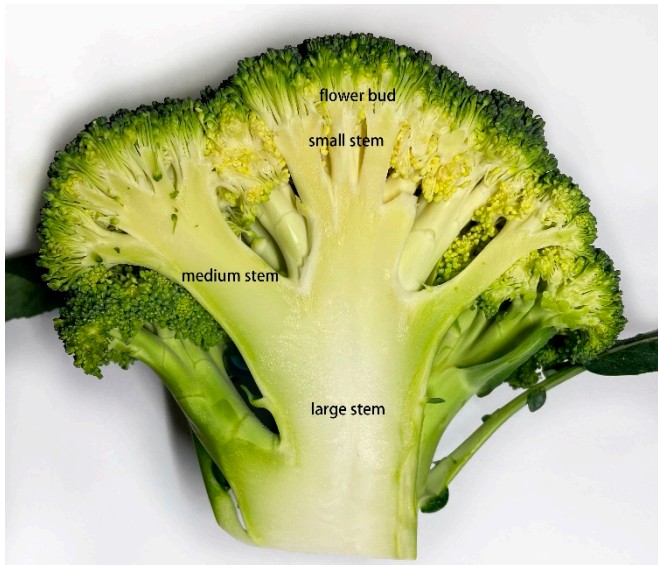

**Figure 1.** Example of a broccoli block considered in this study.

In this study, experiments were performed on large blocks (Figure 2) obtained by primary block-cutting equipment [17]. As shown in Figure 3, most large blocks after primary block cutting and sorting were ellipsoidal, and the maximum size of the blocks in the three directions indicated by dimensions a, b, and c, are 80–90 mm, 50–70 mm and 40–60 mm, respectively, based on measurements and statistical metrics. Broccoli blocks larger than 70 mm are conveyed into the broccoli secondary cutting equipment at an interval of 1 s.

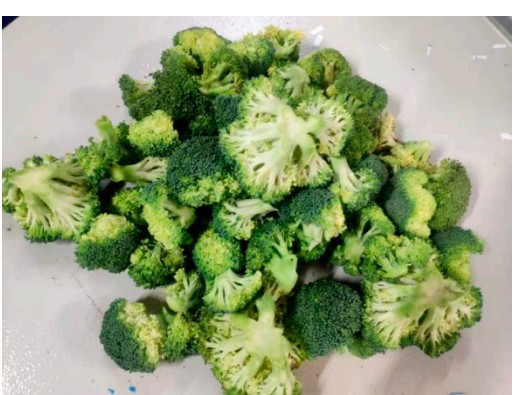

**Figure 2.** Broccoli samples after first cutting.

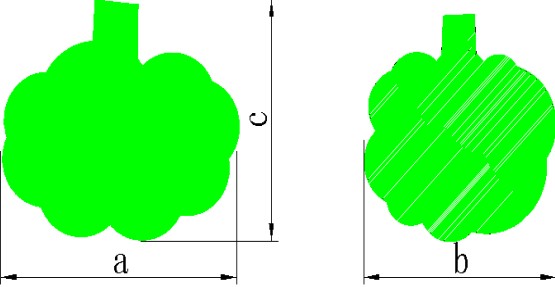

**Figure 3.** Size diagram of broccoli blocks after the first cutting.

### 2.2. Design of Screening Device with Differential Round Belts

#### 2.2.1. Specifications of Screening Device with Differential Round Belts

The blocks obtained from primary cutting are sent to the screening device by a conveyor belt with a tray at a speed of 300 mm/s. To reduce the friction that brings harm to the broccoli, the minimum design speed of the two circular belts is 300 mm/s and a 70 mm gap is left between the belts. As shown in Figure 4 below, by appropriately increasing the speed of one of the round belts, the blocks will rotate between the round belts, and the larger blocks will remain above the two round belts while the smaller blocks will fall through the gap into the collection frame below.

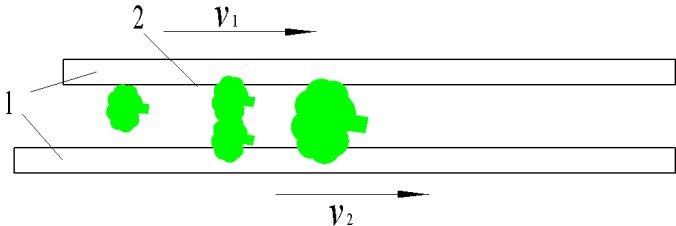

**Figure 4.** Schematic diagram of blocks above differential round belts, where 1. Belts; 2. Blocks.

To meet the minimum line speed of 300 mm/s, the initial design parameters of the differential circular belt conveyor screening device are as follows: the diameter of the belts is 150 mm; the diameter of the circular belt is 18 mm; and the speed of the circular belts min is 40 rpm calculated from the above parameters.

#### 2.2.2. Simulation Analysis of Screening Device with Differential Round Belts Transmission and Screening Motion Based on Solidworks Motion

As shown in Figure 5, with the parameters calculated in the previous subsection, the broccoli block was modeled in a simplified way, replacing the differential belts with a cylindrical bar, and setting stoppers on both sides of the round bar to prevent the block from falling out. In SolidWorks motion, the gravitational direction of the simulation environment is set vertically downward, the material properties are given to the simulation objects, the contact properties of the entities between the simulation objects are set separately, and the high-precision contact is turned on. The linear motor is set with the same direction and different speeds for the two circular belts, respectively, by trajectory tracking the blocks' center of mass as the position to obtain the small blocks' fall position and screening time. The blocks conveyed by the chainplate are disordered, as shown in Figure 6, the simulation analysis is carried out with two kinds of motions—staggered small blocks and stacked between large and small blocks—to obtain the most suitable differential speed pair, and this set of parameters is transported to the screening simulation of a stack of blocks to obtain the screening interval time.

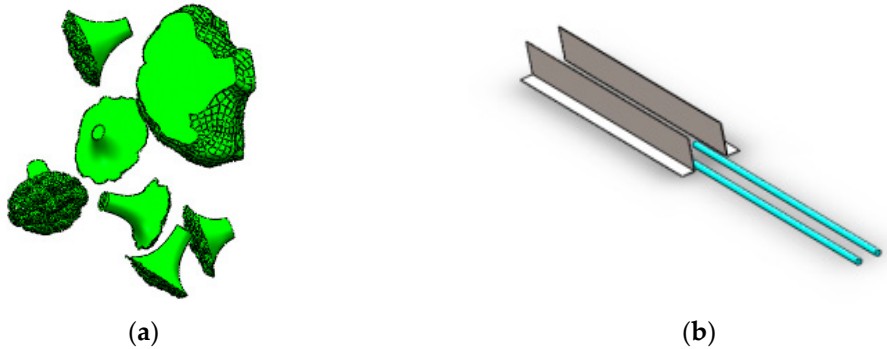

(**a**)  (**b**)

**Figure 5.** Simplified modeling of simulation objects: (**a**) blocks; (**b**) cylindrical bar and stoppers.

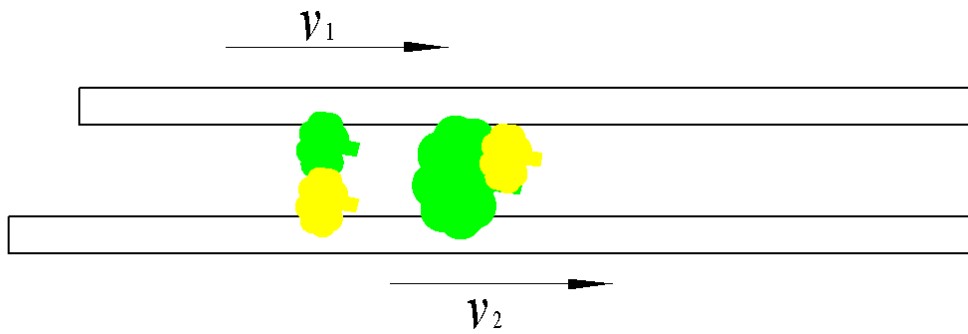

**Figure 6.** Schematic diagram of the two simulation situations: crossed small blocks on the left; stacked between large and small blocks on the right.

First, fix a circular belt speed of 300 mm/s, and set another circular belt speed of 400 mm/s, 500 mm/s, and 600 mm/s in turn (because the initial speed of the block is 300 mm/s, the speed of the other circular belt should not be set too high to avoid damage to the blocks). Shown in Figure 7 for the small block staggered screening simulation analysis charts, respectively, are the trajectory of two blocks and the distance and staggering time of the center of mass of the two blocks moving in the direction of gravity.

**(a)**  **(b)**  **(c)**

**Figure 7.** The small block staggered screening simulation analysis charts, respectively, the trajectory of two blocks and the distance and separation time of the center of mass of the two blocks moving in the direction of gravity: (**a**–**c**) correspond to 300:400, 300:500, and 300:600, respectively.

From the simulation results analysis of small blocks staggering, in the three-speed groups, the differential circular belt can realize the blocks screening so that the small blocks fall from the gap between the two circular belts. The screening times of the three-speed groups are 300:600 mm/s, 300:500 mm/s, and 300:400 mm/s, corresponding to 0.11 s, 0.13 s, and 0.14 s, respectively, and the speed combination of 300 mm/s and 600 mm/s has the shortest screening time, which is most suitable for the screening of blocks in the case of staggering small blocks.

As shown in Figure 8 for the large and small block stacking screening simulation analysis charts, respectively, the trajectory diagram of the two blocks are simulated as well as the center of mass of the two blocks in the direction of gravity to move distance and stagger time.

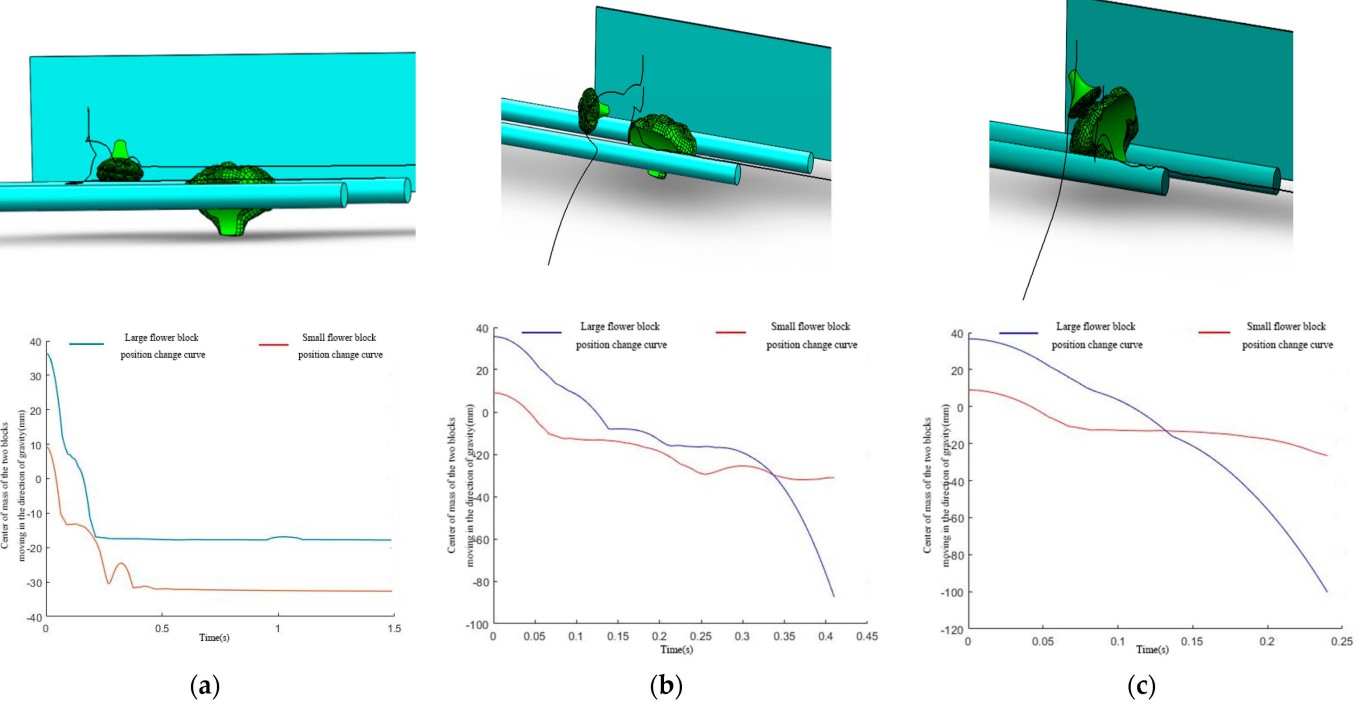

**Figure 8.** Large and small blocks stacked screening simulation analysis figures, respectively, the trajectory of two blocks and the distance and separation time of the center of mass of the two blocks moving in the direction of gravity: (**a–c**) correspond to 300:400, 300:500, and 300:600, respectively.

According to the simulation results above, it can be concluded that:

(1) Under the differential speed ratio of the circular belt conveying line speed of 300 mm/s and 400 mm/s, the position of the center of mass in the gravity direction does not change after 0.5 s, indicating that the blocks are still stacked with each other and are not staggered, so the differential speed ratio has little effect on the screening of the block stacking.

(2) Under the differential speed ratio of the circular belt conveying line speed of 300 mm/s and 500 mm/s, the position of the small block in the direction of gravity is continuously going down and the position of the large block does not change much to achieve separating and screening at time 0.35 s.

(3) Under the differential speed ratio of the circular belt conveying line speed of 300 mm/s and 600 mm/s, the position of the small block in the direction of gravity is continuously going down and the position of the large block does not change much to achieve separating and screening at time 0.13 s, which means it is faster block separation compared to the round belt conveying lines with speeds of 300 mm/s and 500 mm/s.

With the above two simulation results, the differential speed ratio of 300 mm/s and 600 mm/s for the round belt conveying line is more favorable for block separation. Then,

according to this differential speed ratio, after primary block cutting, blocks are transported into the round belt screening device for differential screening simulation as shown in Figure 9 in a bunch of blocks at the same time to the conveyor belt; after a second, the small blocks have all fallen, while the large blocks are still in the differential belt, which shows a better differential screening simulation effect.

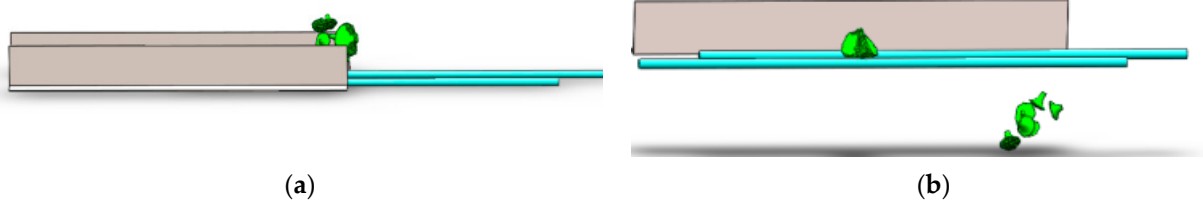

(**a**)                                                                                            (**b**)

**Figure 9.** Blocks stacked screening simulation analysis figures, respectively, the blocks attitude at the beginning of simulation (**a**) and at the end of 2s simulation (**b**).

*2.3. Design and Analysis of Broccoli Secondary Cutting Equipment*

As shown in Figure 10, blocks, which are conveyed one by one by the screening device, whose attitude is adjusted by the double-spiral adjusting mechanism, fall into the double-baffle conveyor belt for delivery one by one and finally are centered by the centering chute so that the disc-type cutter can cut the block along the middle of the longest size of the block to perform the secondary cutting. This process mitigates excessive differences in the size of the florets and increases the success rate of obtaining qualified floret sizes after the second cutting.

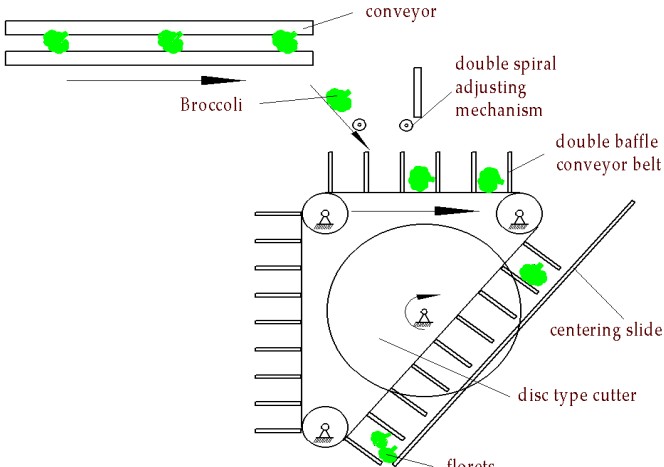

**Figure 10.** Schematic diagram of the broccoli secondary cutting process and equipment.

2.3.1. Design and Principle Analysis of the Double-Spiral Adjusting Mechanism

As shown in Figure 11, the twin-screw extrusion mechanism can be used in the block adjusting mechanism which was renamed as the double-spiral adjusting mechanism [18–20]. The double-baffle conveyor belt is directly driven by the motor that drives the left spiral rod through the primary synchronous belt, and the second synchronous belt with the same tooth ratio drives the right spiral rod. The spiral rod with the opposite direction of rotation rotates unsuitable blocks, and the adjusted blocks fall through the gap between the spiral rods into the conveyor belt with the spacer baffle so that the blocks can be adjusted for centering and conveying. To achieve these effects, instead of causing the blocks to bind at the baffle of the conveyor belt, the following analysis and calculations are performed.

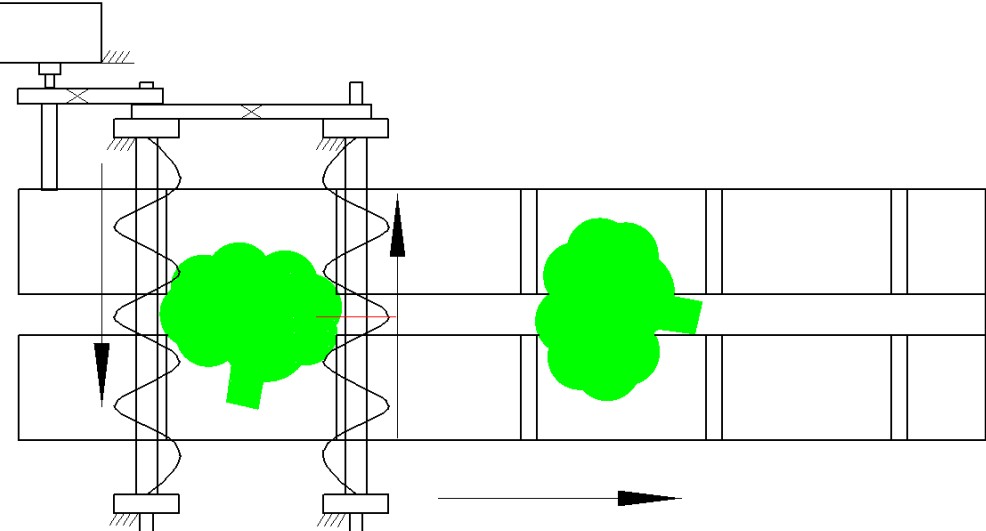

**Figure 11.** Schematic diagram of the operation of the double spiral adjusting mechanism.

According to the dimensions (*b* and *c*) of the blocks, the distance (s) and screw pitch ($P_h$) of the double spiral rods were designed. To ensure the right attitude (i.e., the direction of the longest dimension of the block is nearly perpendicular to the direction of the conveyor of the baffle conveyor belt below, falling between the baffles) of the blocks to fall in the distance between the spiral rods, the designed distance must satisfy:

$$a > s > b\&c \tag{1}$$

After measuring the majority of blocks, dimensions (*b* and *c*) are less than 70 mm, and the dimension (*a*) is 80–90 mm. We thus determine the distance s, which must be 10 mm larger than the largest value among *b*, *c* of the double spiral rod, which is 80 mm. To mitigate block stacking and achieve diversion, the attitude of the blocks must be quickly adjusted. The screw pitch must satisfy:

$$b\&c > P_h \geq \frac{a}{2} \cdot \sin \theta \tag{2}$$

where $\theta$ is the rotation angle of blocks pushed by the screw rod and considered to be 30°. Then, we use $P_h$, which is greater than or equal to 22.5 mm and calculated to be 30 mm. Thus, we determine if one turn of the screw rods will adjust the attitude of the blocks sufficiently.

2.3.2. Motion Analysis and Parameter Calculation of the Double Spiral Adjusting Mechanism Adjusting Block Posture

Because the attitude of the blocks falling from the conveying device onto the spiral adjusting mechanism is random, we determine the spiral rods' rotation speed by analyzing the random attitudes of the blocks and deriving the longest time required to adjust their attitudes.

The double-spiral adjusting mechanism uses trapezoidal spiral rods, as shown in Figure 12, whose sliding speed is:

$$v_L = \frac{n_L P_h}{60} \tag{3}$$

where $v_L$ is the sliding speed, $P_h$ is the screw pitch, and $n_L$ is the rotation speed of the trapezoidal spiral rod.

In the first case, blocks fall to the right (Figure 12a) of the screw rod or the left (Figure 12b) and cannot fall through the double screw rods. As shown in Figure 12, the solid line in the figure is the initial state of the block after falling, and the dashed line is the

state after adjusting the attitude by the propulsion of the screw rod and being able to fall from the two spiral rods.

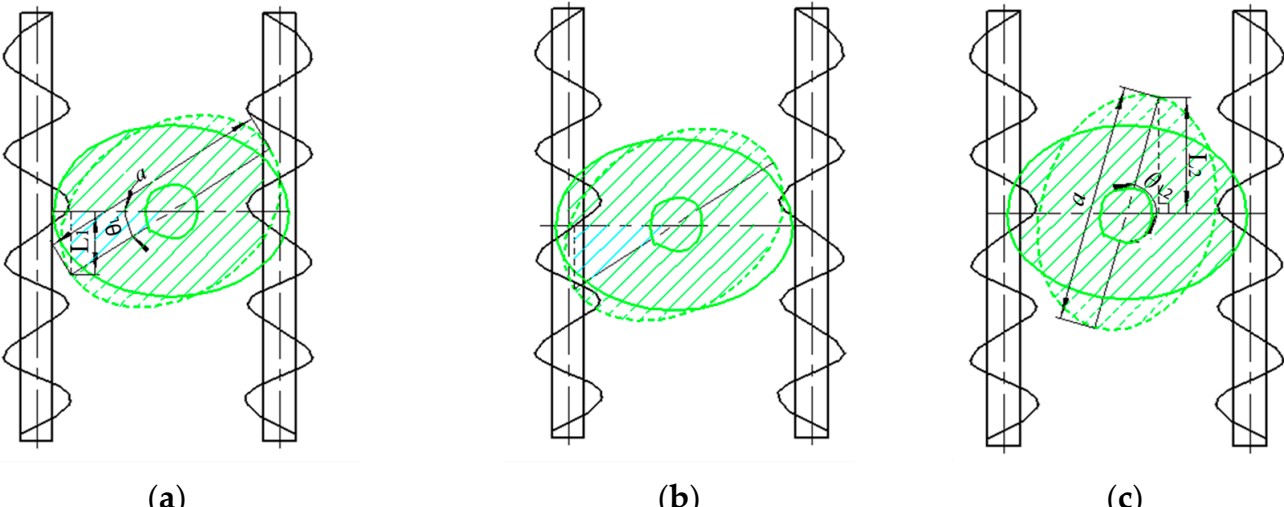

**Figure 12.** Schematic diagram of the two cases of the block falling onto the double spiral adjusting mechanism: (**a**) block biased to the right of the spiral rod; (**b**) block biased to the left of the spiral rod; (**c**) the direction of the longest dimension of the block has a large taper angle or is even perpendicular to the axis of the spiral rod.

When the block in the above state falls onto the baffle conveyor belt, the time required for the screw rod to adjust the block can be expressed as:

$$t_{L_1} = \frac{L_1}{v_L} = \frac{60 \cdot \frac{a}{2} \cdot cos\,\theta_{L_1}}{n_L \cdot P_h} \tag{4}$$

where $t_{L_1}$ is the time required for the spiral rod to adjust the block in the first case and $\theta_{L_1}$ is the angle of block rotation in the first case.

In the second case, the direction of the longest dimension of the block has a large taper angle or is even perpendicular to the axis of the spiral rod, as shown in Figure 12c.

When the block in the second state falls onto the baffle conveyor belt, the time required for the screw rod to adjust the block can be expressed as:

$$t_{L_2} = \frac{L_2}{v_L} = \frac{60 \cdot \frac{a}{2} \cdot cos\,\theta_{L_2}}{n_L \cdot P_h} \tag{5}$$

where $t_{L_2}$ is the time required for the spiral rod to adjust the block in the first case and $\theta_{L_2}$ is the angle of block rotation in the second case.

Comparing these two cases, where $L_2 > L_1$ the screw rod of the second case takes more time to adjust the attitude of the block. According to the attitude adjustment time in the second case, the speed relationship of the spiral rods and the baffle conveyor belt is determined to achieve diversion and mitigate block accumulation.

The blocks are transported by the transfer device at 1 s intervals and fall into the double spiral adjusting mechanism with a flat throwing motion, as shown in Figure 13.

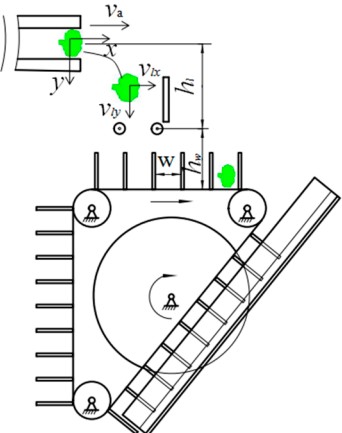

**Figure 13.** Schematic diagram of the process of the blocks falling into the double-baffle conveyor belt.

The time ($t_l$) for the block to move from the conveyor to the spiral rods and the time ($t_s$) for it to fall freely onto the baffle conveyor belt can be expressed, respectively, as:

$$t_l = \sqrt{\frac{2h_l}{g}} \tag{6}$$

$$t_s = \sqrt{\frac{2h_w}{g}} \tag{7}$$

where $t_s$ is the height difference between the conveyor and the double screw rods, and $h_w$ is the height difference between the spiral rods and the baffle conveyor belt.

The time $h_w$ for one pitch of the baffle belt operation can be expressed as:

$$t_w = \frac{60w}{\pi n_w d_w} \tag{8}$$

where $w$ is the interval distance of the baffle conveyor belt, $n_w$ is the rotation speed of the baffle conveyor belt pulley, and $d_w$ is the diameter of the pitch circle of the baffle conveyor belt pulley.

To achieve diversion, the following two conditions must be met: (1) the sum of the interval time of the conveying device and the time of the block from the conveying device to the spiral rod must be greater than the time of the spiral rod adjusting block, and (2) the sum of the time of the spiral rod adjusting block and the time of the block falling from the spiral rods onto the baffle conveyor belt must be greater than the time of the baffle conveyor belt operating an interval. These times can be represented as follows:

$$\begin{cases} t_g + t_l > t_{L_2} \\ t_{L_2} + t_s > t_w \end{cases} \tag{9}$$

$$\begin{cases} t_g + \sqrt{\frac{2h_l}{g}} > \frac{60 \cdot \frac{a}{2} \cdot \cos\theta_{L_2}}{n_L \cdot P_h} \\ \frac{60 \cdot \frac{a}{2} \cdot \cos\theta_{L_2}}{n_L \cdot P_h} + \sqrt{\frac{2h_w}{g}} > \frac{60w}{\pi n_w d_w} \end{cases} \tag{10}$$

where $h_l$ and $h_w$ are determined by the block maximum size statistics, leave a certain margin, and are set equal to 100 mm and 90 mm, respectively; $d_w$ and $w$ are determined according to the block size and selection, and are set equal to 120 mm and 85 mm, respectively; and $\theta_{L_2}$ is obtained according to the size of the block statistics and approximately equals 30°. These values were determined using Equation (9) to calculate the speed of spiral rods ($n_L > 64.5$ rpm) and the speed of baffle conveyor plate ($n_w > 10.61$ rpm); thus, the above conditions are satisfied to ensure block diversion and avoid accumulation.

### 2.4. Design and Parameter Calculation of the Double-baffle Conveyor Belt

In this study, a double isometric baffle conveyor belt solution is used, where the space between the baffles can accommodate an individual block and convey it individually.

As shown in Figure 14, the double-equidistant-baffle conveyor belt is constructed in an inverted right triangle so that the centering device and the second cutter for the block can be laid out at its bevel.

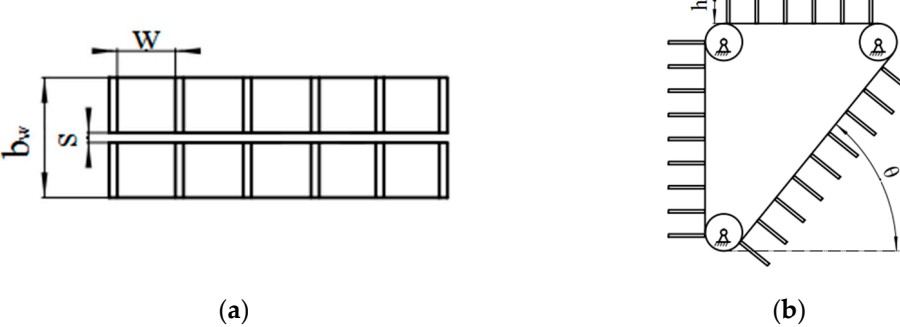

(a)                                        (b)

**Figure 14.** Schematic diagram of the double-equidistant-baffle conveyor belt: (**a**) top view; (**b**) primary view.

The primary parameters of the double-equidistant-baffle conveyor belt are selected by considering the size of the block (see Section 2.1) and the layout of the centering chute and disc-type cutter. To make it easier for the blocks to fall on the conveyor belt and not fall outside, the total width $b_w$ of the double-equidistant-baffle conveyor belt is designed to be 100 mm, the interval $\theta$ between the two baffles is 75 mm, and the height of the baffle $h_w$ is 80 mm. A disc-type cutter with a thickness of 2.5 mm is installed in the gap (S = 6 mm) between the two baffle conveyor belts to avoid interference.

### 2.5. Design and Parameter Calculation of the Centering Chute

The double-baffle conveyor belt of Section 2.3 is designed according to the maximum size of each block, which can be different and may cause the block to be biased to the left or right on the baffle conveyor belt, resulting in a situation where it is not centered for conveying. Thus, the following section describes the centering chute, which adjusts the position of a block that must be centered.

As shown in Figure 15, the inner side of the centering chute has two inclined structures that center blocks in the middle of the inclined structure, while the through-hole in the middle of the chute allows a portion of the disc knife to pass through the chute.

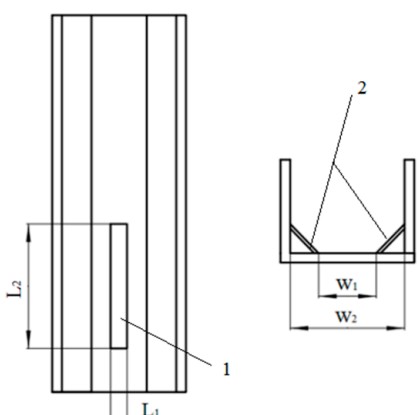

**Figure 15.** Schematic diagram of the process of the blocks falling into the double-baffle conveyor belt, where 1. through-hole; 2. inclined structures.

According to the size of the block and the width of the baffle plate, the minimum width of the inclined structure of the slide is 80 mm, and the distance between the baffle plates of the slide is 106 mm to avoid interference between the baffle conveyor belt and the slide. The width and length of the through-hole are 5 mm and 110 mm, respectively, depending on the diameter (D = 250 mm) and thickness ($w_c$ = 2.5 mm) of the disc cutter and the relative position of the cutter and the chute. To make the double-equidistant-baffle conveyor belt and the slide not interfere, the distance is set equal to 4 mm.

As the block moves through the centering chute, the inclination of the centering chute must be sufficiently high to have the gravitational force in the direction of the centering chute be higher than the sliding friction, which will keep the blocks from being pushed by the double-baffle conveyor belt and the centering chute extrusion, causing damage:

$$mg \sin \theta > \mu_t mg \cos \theta \tag{11}$$

The material of the slide is PLA. Taking the friction coefficient $\mu_t$ as 0.51 and substituting it into Formula (11), we can calculate the inclination angle $\theta$ as > 27.02°; we thus select 30° for $\theta$ to reduce damage to the block during conveying.

### 2.6. Analysis of the Force on the Block during the Second Cutting Process and Disc Cutter Design

After attitude and centering adjustments, each block is supported by the baffle below and moves along the chute to the second cutting position. The disc cutter is arranged at the oblique top of the slide groove and is driven clockwise by the motor. The block slides down to the disc cutter under the combined force of gravity and friction and is held in place on contact by pushing the back baffle until it is cut in half by the disc cutter. The maximum cutting force $F_q$ required for broccoli is 15 N according to the broccoli cutting technology study published by Chen Liqun et al., [10]. This force dictates the required motor torque (*T*):

$$T = \frac{F_q D}{2} \tag{12}$$

where *D* the diameter of the disc cutter is considered to be 250 mm. The minimum value of the torque is 1.87 N·m. Therefore, a common geared motor with a torque of 3 N·m is used to obtain a cutting force of 24 N, which satisfies the cutting force requirement.

## 3. Results

### 3.1. Broccoli Secondary Cutting Equipment and Operation Process

Using SolidWorks software, according to the parameters obtained from the above chapters, the selection design mechanism part size and relative position are used to complete the 3D virtual prototype assembly and the parts processing assembly and commissioning, as shown in Figure 16. Spiral attitude-adjusting mechanism, double-baffle conveyor belt, block-centering chute, and disc-type cutter are fixed on the cross beam of the frame from top to bottom. The screening mechanism is also fixed to the frame in front of them, and its output port is over the twin-screw mechanism. The output shafts of the two motors are connected to the belts shafts at 25 rpm and 50 rpm, respectively, to achieve a differential speed ratio of 300 mm/s and 600 mm/s. Both sides of the round belts are fixed with baffles to prevent the broccoli from rolling out after falling due to untimely screening. The distance between the two round belts with a diameter of 18 mm is kept constant at 70 mm by controlling the shoulder length of the belts shafts. The twin screws in the spiral attitude-adjusting mechanism are fixed on the frame by the bearing seat through the T-bolt according to the distance s (80 mm). The three apex angles of the right triangle double-baffle conveyor belt, which is according to the spacing w (85 mm) evenly distributed baffle, are provided with synchronous belt wheels, which are fixed by the bearing seat on the frame, respectively, and keep the spacing S (6 mm) between double conveyor belt. One right Angle edge of the double-baffle conveyor belt is parallel to the double helix mechanism to ensure that the blocks that have been adjusted can be caught reposefully. The block-centering

chute is fixed to the frame by slots on both sides and is parallel to the long side of the double-baffle conveyor, and the collection frame is placed under the end of the chute to receive the second cut blocks. Inside the double-baffle, the conveyor belt is installed with a disc-type cutter, which is fixed on the drive shaft by the flange, and the disc-type cutter rotation is guaranteed to be stable by the bearing seat on both sides. While the distance between the center of the tool and the center slide of the block is less than the radius of the tool to ensure that the disc-type cutter can completely cut blocks. The overall power of the mechanism comes from the two motors on one side. The conveyor motor (6IK180RGN-CF, Matsuoka transmission) is speed-controlled to 11 rpm and connected to the input shaft of the double-equidistant-baffle conveyor belt, which drives the double spiral rod via a synchronous belt with a ratio of 6:1. The tool motor (4IK25RA-C, Matsuoka transmission) is speed-controlled to 83 rpm, and the torque is 3 N m to drive the rotation of the disc tool.

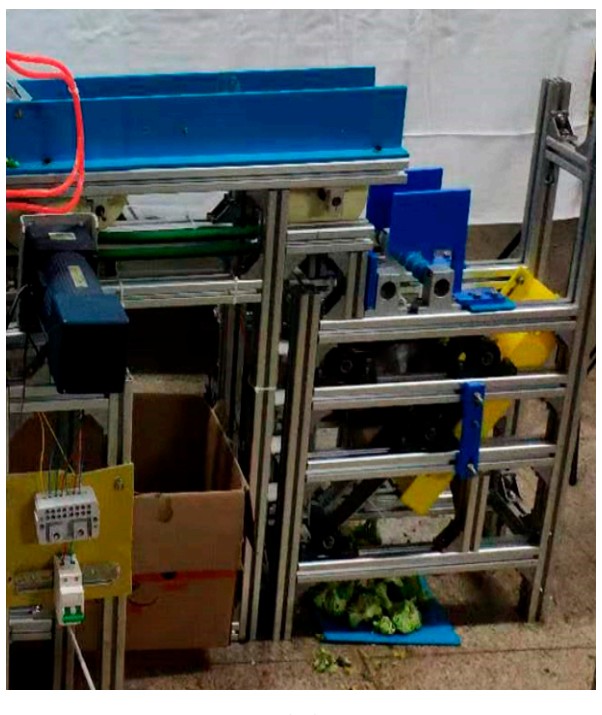

(**a**)

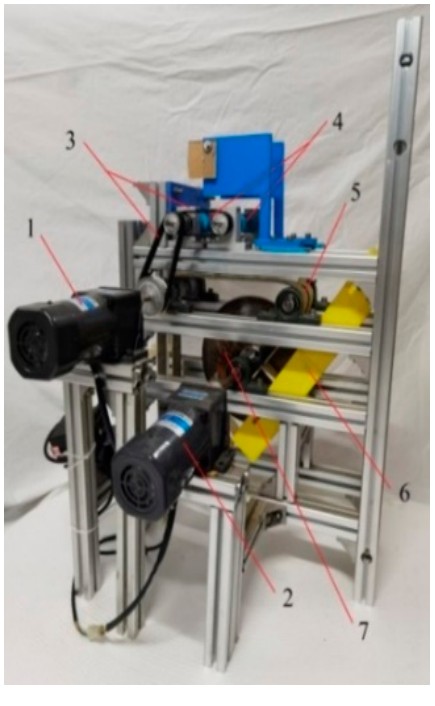

(**b**)

**Figure 16.** Equipment assembly diagram: (**a**) Physical picture of secondary cutting overall equipment including screening mechanism and spiral attitude-adjusting mechanism, double-baffle conveyor belt, block-centering chute, and disc-type cutter; (**b**) Physical picture in another perspective of secondary cutting overall equipment without screening mechanism, where 1. Delivery motor; 2. Tool motor; 3. Drive Belt; 4. Double spiral rod; 5. Baffle conveyor belt; 6. Centering chute; and 7. Disc cutter.

Test process: Screening mechanism motors startup. The delivery motor begins, driving the baffle conveyor belt and double spiral rod to move, while the cutter motor drives the disc cutter. The blocks are delivered one by one by the conveyor to the double spiral adjusting mechanism, which adjusts each block's attitude. Each block then falls into the double-baffle conveyor belt for transport and is centered by the centering chute so that the disc-type cutter can cut the block along the middle of its longest size to complete secondary cutting. The control flow is shown in Figure 17.

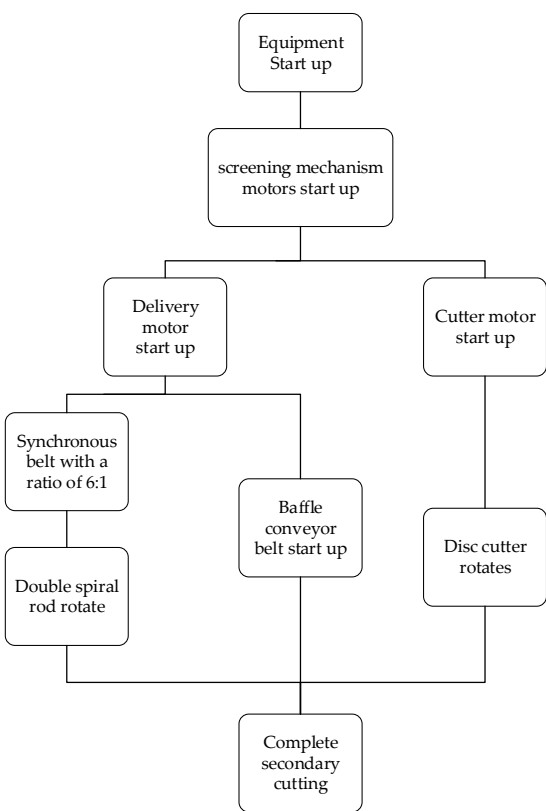

**Figure 17.** Control flow chart of broccoli secondary cutting equipment.

*3.2. Results and Analysis of Experiment*

In this study, the primary cutting and screening equipment shown in Figure 18 was used to perform the first cutting test on broccoli to obtain blocks, and large blocks with a cut size greater than 70 mm were screened out by the screening device with differential round belts. The second cutting was performed as described in the previous section.

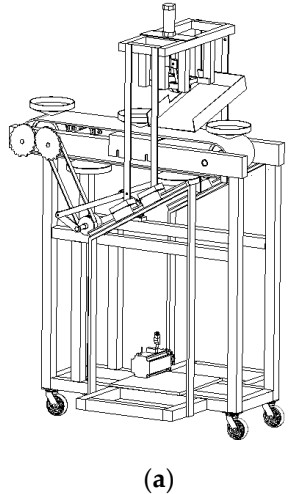

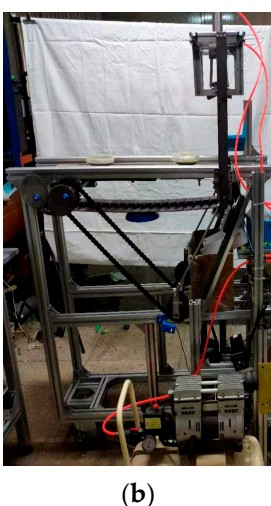

(**a**)                                                         (**b**)

**Figure 18.** Primary cutting equipment assembly diagram; (**a**) Three-dimensional assembly drawings; (**b**) Photograph.

The following tests will be carried out to verify the feasibility of the above simulation and the practicality of the mechanism in screening and secondary block cutting for the differential belts screening mechanism and secondary block cutting mechanism. The test

set three groups of round belt conveying line speed: 300 mm/s and 400 mm/s; 300 mm/s and 500 mm/s; and 300 mm/s and 600 mm/s. Each group randomly took 20 broccolis, three groups of a total of 60 broccolis for the test. After primary cutting, the blocks from the conveyor chain plate into the differential circular belts and the video of differential screening were recorded using a high-speed camera of the Phantom series, and the following Figure 19 was captured by the image analysis software.

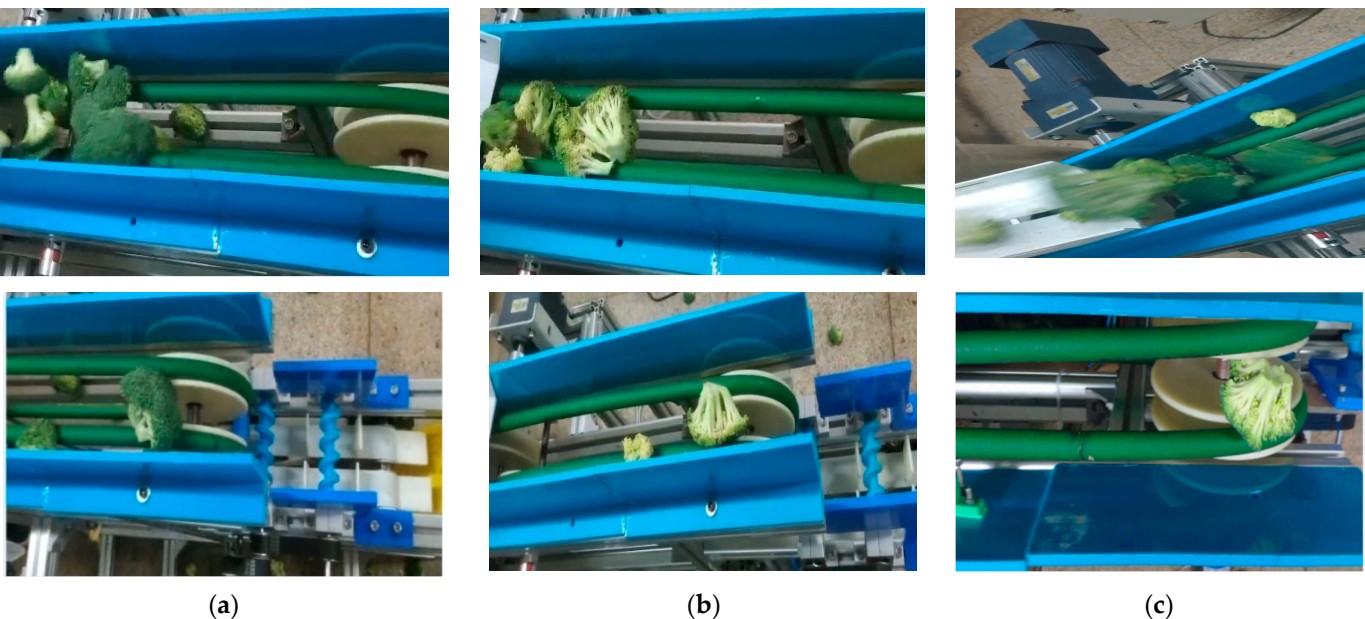

(**a**)    (**b**)    (**c**)

**Figure 19.** Large and small blocks stacked screening simulation analysis figures, respectively, the trajectory of two blocks and the distance and separation time of the center of mass of the two blocks moving in the direction of gravity: (**a–c**) correspond to 300:400, 300:500, and 300:600, respectively.

The total number of blocks, florets, and unscreened florets were counted, and the screening success rate was defined as the ratio of the number of florets obtained by screening to the total number of florets, and the results of the experiment were recorded, as shown in Table 1

**Table 1.** Screening results of blocks on differential speed belts with different speed ratios.

| Differential Belt Ratio (mm/s) | Blocks | Florets | Unscreened Florets | Screening Success Rate |
|---|---|---|---|---|
| 300:400 | 158 | 136 | 12 | 91.2% |
| 300:500 | 168 | 142 | 8 | 94.4% |
| 300:600 | 160 | 137 | 3 | 97.8% |

As shown in Figure 19 and Table 1, it can be seen that the screening effect is similar to the simulation results, the 300 mm/s and 400 mm/s groups were not completely screened, there are still some small blocks left on the round belt or in the gap between the side baffles and the round belts, this situation occurs more or less in the remaining two groups, only the probability of occurrence is different. Each group of 20 blocks obtained a slightly different total number of blocks after the first cut, and the number of small blocks thus varied somewhat. The screening success rate after different differential speed ratio tests was significantly different, but the screening success rate of the round belt line speed of 300 mm/s and 600 mm/s, respectively, was 97.8%, which was the highest screening success rate among the three groups of differential speed ratios and was more favorable to the separation of the stacked blocks and the screening of the small blocks.

The larger blocks screened by the screening mechanism go through the secondary cutting mechanism one by one. the florets were collected, as shown in Figure 20a. After primary cutting, 117 blocks, in which the maximum size of the blocks inthree directions was a, were obtained, and the second cutting cut blocks into two small florets, which measured at the maximum size. The larger the value $\alpha_{lager}$, the smaller $\alpha_{smaller}$, as shown in Figure 21 which demonstrates the data of the florets after the second cut.

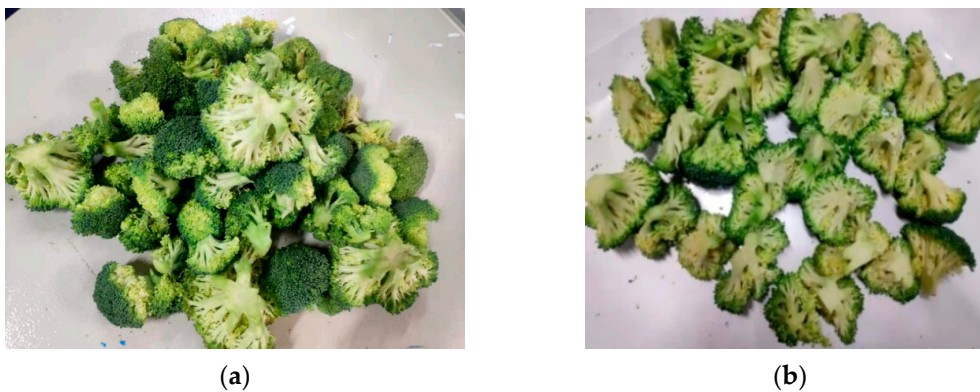

(**a**)  (**b**)

**Figure 20.** Changes in broccoli block morphology during the broccoli cutting process; (**a**) broccoli after the primary cutting; (**b**) broccoli after the second cutting.

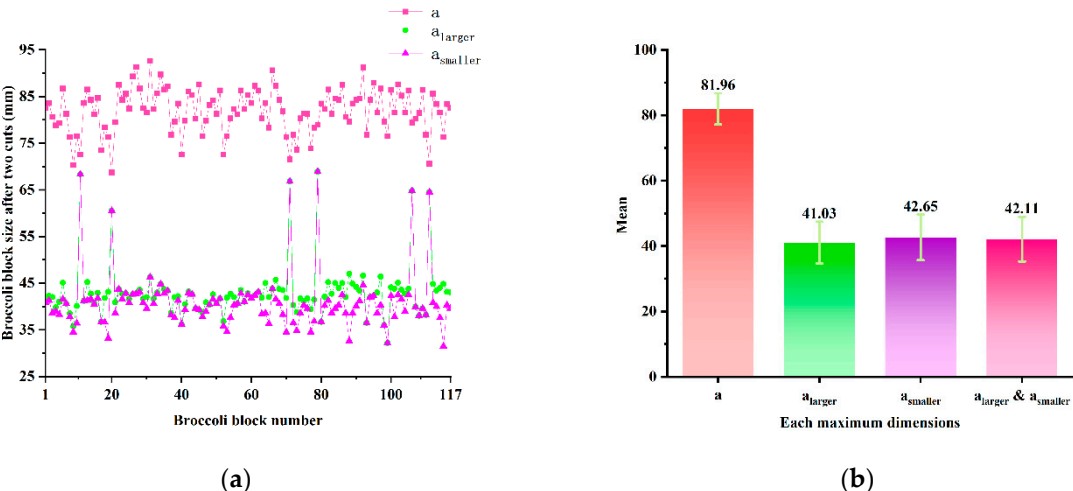

(**a**)  (**b**)

**Figure 21.** Comparison diagram of the maximum size of broccoli block: (**a**) Maximum size of each block after two cuts; (**b**) Average value and error of the maximum size of the block after two cuts.

We counted the results of 117 large blocks screened by the screening mechanism at a differential speed ratio of 300 mm/s:600 mm/s to complete the secondary cut in the next mechanism. A total of 234 florets were obtained after secondary cutting. Six blocks were not cut in the direction of the maximum size $\alpha$ during the second cut due to the block not being adjusted to the correct attitude. We measured $\alpha_{lager}$ and $\alpha_{smaller}$ to be similar to $\alpha$; conversely, the remaining 111 blocks, whose maximum size was over 70 mm, were correctly cut to be less than 50 mm. The time and count of the secondary block-cutting equipment were used to calculate that the proposed equipment can cut 47 blocks per minute. For the 222 florets obtained after secondary cutting, their size was within 70 mm, which meets market size requirements. The size difference of the correspondingtwo florets' maximum sizes was 0–8 mm, the mean value of the maximum size of the florets was 42.65 mm, and the variance was 48.9 mm. The variance is slightly larger because the difference in the size of each block obtained from the primary block cutting increases the size of the block obtained after secondary cutting. However, in contrast, the difference in the mean maximum size of

the secondary-cut blocks was only 3.24 mm, which confirms the reliability of the proposed secondary-cut block mechanism. These results show that the proposed double-spiral adjusting mechanism and centering chute scheme are feasible. Comparing our device with existing devices, as shown in Table 2, the D-Core 30i/50i can achieve either 30 or 50 cuts per minute. Liqun Chen's device was able to complete 50–60 broccoli cuts in one minute with a 91% success rate, while the device proposed in this paper was able to achieve 47 cuts in one minute with a 94.8% success rate. It can be concluded that the equipment achieved the level with the existing primary block-cutting equipment, and roughly met the rate of primary block cutting, which can be used with the primary block-cutting equipment.

**Table 2.** Comparison of block-cutting efficiency and success rate of different equipment.

| Reference | Success Rate | Efficiency |
|---|---|---|
| This work | 94.8% | 47 per minute |
| D-Core 30i/50i | / | 30/50 per minute |
| Chen | 91% | 50–60 per minute |

## 4. Discussion

From the characteristics of past equipment for cutting broccoli, it can be found that the problem of a single cutting is that there is a large gap between the size of the blocks, and the existence of large-sized blocks does not meet the market demand. In such a context, this paper proposes the use of the screening device, spiral attitude-adjusting mechanism, double-baffle conveyor belt, block centering chute, and disc-type cutter: five components to achieve. The blocks are sequentially screened by the screening device to obtain large blocks, posture-adjusted by the spiral attitude-adjusting mechanism, transported by the double stopper and centering chute to the circular tool to complete the secondary cutting. To realize the above process, the design mechanism is proposed in this paper, and the simulation analysis is carried out for the scenarios of staggered small blocks and stacked between large and small blocks using different differential ratios, and the most suitable differential ratio of 300 mm/s:600 mm/s is obtained. By analyzing the two cases of blocks on the spiral attitude-adjusting mechanism, the suitable speed of the spiral rod and the speed of the double-baffle conveyor belt are 64.5 rpm and 10.61 rpm, respectively. To reduce the damage in the process of block conveying and the inclination, the angle of the inclined centering chute was calculated to be greater than 27.02°. The parameters of the blade and its driving motor were calculated.

Finally, the paper combines the simulation analysis with the calculation results, builds a 3D model of the device, and prepares a physical prototype for testing to verify the correctness of the above analysis and the practicality of the device in a real secondary cutting scenario. Each group randomly took 20 broccolis, three groups of a total of 60 broccolis for the test. After primary cutting, the blocks from the conveyor chain plate were transported to into the differential circular belts. The total number of blocks, florets, and unscreened florets were counted, and the results of the comparison test were calculated to show that a differential speed of 300 mm/s:600 mm/s achieved 97.8% of the florets being screened. The results corroborate the previous simulation analysis that this differential belt scheme is suitable for broccoli screening at the right differential ratio. We completed cutting experiments while counting the results of 117 large blocks screened by the screening mechanism at a differential speed ratio of 300 mm/s:600 mm/s to complete the secondary cut in the next mechanism, in which we measured the florets produced. The size difference among the small florets ranged from 0 to 8 mm, the mean value was 42.65 mm, the maximum size difference was 3.24 mm, the variance was 48.9 mm, the success rate of cutting was 94.8%, and the efficiency was 47 pieces/min, which verified the rationality and feasibility of the second cutting equipment plan and were compared with existing devices. It can be concluded that the equipment achieved the level with the existing primary block-cutting equipment, and roughly met the rate of primary block cutting; the secondary cutting equipment can be used with the primary block-cutting equipment. Comparing our

device with existing devices, the D-Core 30i/50i can achieve either 30 or 50 cuts per minute. Liqun Chen's device was able to complete 50–60 broccoli cuts in one minute with a 91% success rate, while the device proposed in this paper was able to achieve 47 cuts in one minute with a 94.8% success rate. It can be concluded that the equipment achieved the level with the existing primary block-cutting equipment, and roughly met the rate of primary block cutting, which can be used with the primary block-cutting equipment. Therefore, the proposed equipment may be able to promote the field of broccoli storage and processing, facilitate broccoli storage, reduce food waste, and will be to reduce some of these costs. Of course, the equipment mentioned in the paper has many defects, for example, there is no reasonable layout of the collection frame, which will cause re-damage to the broccoli that is originally intact from the screening and cutting block to obtain the florets. On the other hand, the power source is too much, the whole equipment uses four motors, especially for the screening mechanism, but of course, a wheel system or another way to achieve a 1:2 ratio output can be used. In future research, these two aspects can be expanded to make the machine even better. In addition, the secondary cutting block can also be extended to realize the development of the whole equipment from the primary cutting block to the secondary cutting block to complete the whole process of broccoli cutting block processing.

## 5. Conclusions

Considering the problem of broccoli storage and waste, broccoli cutting equipment was created. As mentioned in the introduction, most of the machines have the common problem that the size of the cut blocks varies, and there are large-sized blocks that do not meet the market demand. The secondary cutting equipment proposed in this paper can complete the secondary cutting by screening, transferring, conveying, and cutting the blocks cut by the primary cutting process. The equipment achieves a 97.8% screening success rate and 94.8% secondary cutting success rate, and the maximum size difference of small florets is only 3.24 mm to meet the market demand. The proposal of this equipment may be able to promote the field of broccoli storage and processing, facilitate broccoli storage, and reduce food waste. Secondary cutting equipment not only solves practical engineering problems but also has important significance in academic aspects. The simulation and computational ideas mentioned in this paper can also be applied to other crops to solve processing problems similar to the same bulk mushrooms and cabbage with a thick main stem. The shortcomings of the equipment in terms of collection and movement sources are also pointed out, and the future outlook in terms of the whole machine research is made [7].

**Author Contributions:** Conceptualization, J.J. and J.C.; methodology, J.J., L.C. and T.C.; investigation, J.J. and R.H.; data curation, R.H. and T.C.; formal analysis, J.J. and J.C.; writing—original draft, R.H.; writ-ing—review and editing, J.J., L.C., T.C. and J.C.; project administration, L.C. and J.C.; funding acquisition, L.C. and J.C. All authors have read and agreed to the published version of the manuscript.

**Funding:** This research was funded by the National Natural Science Foundation of China (Grant No. 51975536) and Basic Public Welfare Research Projects of Huzhou city, Zhejiang Province (Grant No. 2020GZ06).

**Data Availability Statement:** The data presented in this study are available on request from the corresponding author.

**Acknowledgments:** The author would like to thank the Key Laboratory of Transplanting Equipment and Technology of Zhejiang Province for its support.

**Conflicts of Interest:** The authors declare no conflict of interest.

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
