# Peer review of "Design of and Experiment with Secondary Cutting Equipment for Broccoli"

_agriculture, doi:10.3390/agriculture12050650_

Round 1
Reviewer 1 Report
Some minor mistakes should be corrected, as below.
Line 122: The figure number “Figure 2” should be “Figure 6”;
Line 153: “the time hw” should be “the time tw”;
Line 263: The expression "We measured a1 and a2 to be similar to a" should be made clearer in meaning.
Reviewer 2 Report
The authors in their work presented a device for cutting broccoli. It is an interesting but niche site. The work was carried out correctly in terms of content. It would be worth adding a broader description of the structure. According to the editorial comments, Table 1 is outside the scope of the print.
Reviewer 3 Report
The work deals with quite interesting issues regarding the study of design of and experiment with secondary cutting equipment for broccoli.
However, the revision is still needed.
1. The introduction is very narrow. The scientific problem and sufficient background is explained in very short form. The novelty is not highlighted enough.
- The article is more like design work for new equipment.
- The discussion of the results should be more analysis. There is no sufficient analysis of the given results. It is necessary to widen the analysis of results. The given results should be compared with the other authors’ studies in the Result and Discussion section.
- Conclusions not found.
Round 2
Reviewer 3 Report
no commets